# UPLC Technique in Pharmacy—An Important Tool of the Modern Analyst

Paweł Gumułka [1,2], Joanna Żandarek [1,2], Monika Dąbrowska [2,*] and Małgorzata Starek [2,*]

1   Doctoral School of Medical and Health Sciences, Jagiellonian University Medical College, 16 Łazarza St., 31-530 Kraków, Poland
2   Department of Inorganic and Analytical Chemistry, Faculty of Pharmacy, Jagiellonian University Medical College, 9 Medyczna St., 30-688 Kraków, Poland
*   Correspondence: monika.1.dabrowska@uj.edu.pl (M.D.); m.starek@uj.edu.pl (M.S.)

**Abstract:** In recent years, ultra-efficient liquid chromatography (UPLC) has gained particular popularity due to the possibility of faster separation of small molecules. This technique, used to separate the ingredients present in multi-component mixtures, has found application in many fields, such as chemistry, pharmacy, food, and biochemistry. It is an important tool in both research and production. UPLC created new possibilities for analytical separation without reducing the quality of the obtained results. This technique is therefore a milestone in liquid chromatography. Thanks to the increased resolution, new analytical procedures, in many cases, based on existing methods, are being developed, eliminating the need for re-analysis. Researchers are trying to modify and transfer the analytical conditions from the commonly used HPLC method to UPLC. This topic may be of strategic importance in the analysis of medicinal substances. The information contained in this manuscript indicates the importance of the UPLC technique in drug analysis. The information gathered highlights the importance of selecting the appropriate drug control tools. We focused on drugs commonly used in medicine that belong to various pharmacological groups. Rational prescribing based on clinical pharmacology is essential if the right drug is to be administered to the right patient at the right time. The presented data is to assist the analyst in the field of broadly understood quality control, which is very important, especially for human health and treatment. This manuscript shows that the UPLC technique is now an increasingly used tool for assessing the quality of drugs and determining the identity and content of active substances. It also allows the monitoring of active substances and finished products during their processing and storage.

**Keywords:** UPLC; drug analysis; pharmaceutical preparations; quality control; analysis of bioactive compounds; separation techniques

## 1. Introduction

Over the years, the high-performance liquid chromatography (HPLC) technique has gained immense popularity in most analytical laboratories. The liquid chromatograph is believed to be the third most popular laboratory equipment, right after balance and pH meters. Of course, as with any technique, it is constantly being improved. At the beginning of the 1970s, columns (filled with non-porous, irregularly shaped silicate gel of about 40 μm in size) with very low efficiency (the number of theoretical plates was about 1000 per 1 m bed) were commercially available. Later, columns with a grain of 10 μm in diameter were produced, followed by silica gel columns with spherical, porous grains with a diameter of 5 μm. These increased the yield to about 12,000 theoretical plates with a column length of 150 mm. In the 1990s, columns with grains of 3 μm in diameter were created. Subsequently, it was found that further grain reduction was not justified due to the costs and problems associated with their use. The breakthrough year was 2004 when a completely new model of the Waters UPLC liquid chromatograph equipped with columns

with a grain diameter of 1.7 μm appeared on the market. It can now be concluded that UPLC (e.g., ultra-performance liquid chromatography) has proven to be a milestone in liquid chromatography [1]. This technique, serving to separate the components present in mixtures, has found application particular to the analysis of thermally labile or low-volatile compounds. In recent years, it has gained particular popularity due to the possibility of faster separation of small molecules [2]. Chromatographic columns with particles <2 μm are used here, applied in equipment capable of working under high pressure. The flow rates are lower than in classical HPLC, but due to the increase in yield, the total separation time is shortened. This allows the particles to be separated quickly with high efficiency. UPLC is therefore an effective chromatography technique that offers a wide flow range and significantly reduces analysis time.

The basic principle upon which UPLC is based is that as the size of the fill particles decreases, so does the efficiency and hence the resolution. After particle size reduction to less than 2 μm, the efficiency shows a significant increase and does not decrease at increased line velocities or flow rates, in accordance with the van Deemter equation. It is known that the smaller the grain diameter of the column packing, the lower the height of the theoretical plate, i.e., the higher the column efficiency, will be. The minimum of the van Deemter curve corresponds to the ideal flow velocity at which the highest column efficiency is obtained [3]. Thanks to the use of smaller particles, the speed of analysis and the so-called peak capacity (number of peaks per unit time) tend to maximum values. In addition, to improve the efficiency, an increased temperature range should be used (this increases the flow rate of the mobile phase by reducing its viscosity, i.e., significantly lowering the back pressure) and monolithic columns (consisting of a solid piece with flow paths connected by skeletons, so-called passage pores).

The remaining components of the van Deemter equation depend on the grain size. Smaller grains reduce the height, i.e., the column has more theoretical plates per unit length (it is more efficient). Due to the small grains, the analyte can migrate faster to/from the grain as its diffusion path is shorter. This elutes the analyte as a narrow peak (spends less time in the stationary phase where the bandwidth is extended). For example, assuming the grain size will decrease from 5 to 1.7 μm with a constant column length, the resolution should improve 1.7 times, the analysis should be 3 times shorter, the sensitivity will increase 1.7 times, and the pressure will be 27 times higher [4]. However, assuming a constant column length to grain size ratio, the resolution will not change, the analysis will be 9 times shorter, the sensitivity 3 times higher, and the pressure 9 times higher. These dependencies are the reason why producers are constantly working to reduce grain size.

As already mentioned, the separation efficiency increases as the particle size decreases. With smaller particles, the pressure in the column increases significantly, resulting in very high pressures in longer columns. Thus, 1.9 μm columns of the same length as normal 5 μm HPLC columns cannot be used for standard LC systems. For this reason, UPLC columns have lower or similar yields than standard HPLC columns. This translates into faster analysis time but not always better performance.

The main advantages of the UPLC technique include the reduction of analysis time and increased sensitivity and resolution. These changes became possible thanks to the new design of chromatograph elements, including columns, pumps, dozers, and detectors with a reduced volume of measuring cells. The use of short columns and their low packing (1.7 μm) significantly shortened the analysis time. Small column packing forced the use of high pressures (about 1200 bar), and heating the column lowers the viscosity of the solutions, thus increasing the sample flow rate through the system. These changes make it possible to obtain very fast measurement cycle times while maximizing efficiency, which results in a reduction of the dead volume and a shorter stabilization time of the system [5]. The reduction in time reduced analysis costs through more efficient use of the equipment and reduced solvent consumption. At the same time, the increased efficiency of the system allows more information to be obtained than in HPLC.

In theory, the transition from classic to modern UPLC systems should be quick and easy. However, some problems do arise, and training is needed to avoid mistakes, such as those related to other software or with the selection of the appropriate columns. Currently, practically every manufacturer of chromatographic equipment offers equipment capable of working with the pressure required by columns with grains with a diameter of less than 2 μm. Initially, only C18 columns were available, while today, almost all modifications to silica are offered, i.e., C8, phenyl, HILIC, silica, amide, fluoro-phenyl, and phenyl-hexyl. The development was initiated by BEH technology (Ethylene Bridged Hybrid, columns with silica packing reinforced with ethylene bridges), thanks to which the columns are able to operate at a pressure of 15,000 psi (approx. 1000 bar). Another version is High-Strength Silica (HSS) fillings, which are useful in the determination of polar analytes. Unfortunately, they show lower resistance to high pH but, at the same time, higher retention. The latest type of filling is CSH (Charged Surface Hybrid). These are modifications of the BEH columns by giving the surface of the additional charge. Thanks to this procedure, columns filled with these beds (C18, fluoro-phenyl, and phenyl-hexyl) have a wide range of selectivity and make it possible to analyze alkaline compounds tested in acidic phases with low ionic strength (e.g., 0.1% formic acid).

In addition to speed, it is important to increase the resolution. Sample complexity is a huge problem when working with multi-component drug products or molecules with multiple chiral centers. The key to success in such cases is new methods, i.e., apparatus conditions that allow for quick changes of columns or mobile phases. In this case, the use of smaller columns with quick system balancing and the possibility of simultaneous measurement of several quality parameters is an important advantage of UPLC.

The introduction of the UPLC technique created new possibilities for analytical separation without reducing the quality of the obtained results. Many experts have argued that UPLC will replace conventional HPLC techniques. Unfortunately, one of the major disadvantages of UPLC is the financial factor. These expensive devices are not available in all laboratories, and not every researcher will be able to reproduce a given method in his laboratory. Another problem is column padding. When transferring a method from HPLC to UPLC, it is advisable to use the same type of packing. Unfortunately, many existing HPLC fillings are not available in the UPLC version. Moreover, UPLC operates at very high pressures, and the lifetime of the used columns is shortened. Another problem is some aggressive, non-polar solvents that are incompatible with these devices, making it impossible, for example, to separate inorganic ions and polysaccharides.

A very important element of an efficient UPLC system is the selection of the detector. Depending on its type, the sensitivity of the method may increase two to three times in relation to HPLC [6]. Optical detectors based on absorbance, tunable UV/visible detectors, fluorimetric and mass spectroscopy (MS) detectors, etc. are generally used with HPLC. The features of UPLC (i.e., speed, resolution, and sensitivity) make it best suited for use with a mass spectrometer. For MS analyses, source ionization is more efficient with UPLC due to increased peak concentrations with reduced chromatographic dispersion at lower flow rates [7]. The profitability of using the UPLC–MS apparatus makes it a practical tool in the laboratory. This applies in particular to the possibility of working at low flows (on columns with a diameter of 1.0 mm) and the possibility of avoiding flux split, which is a very good tool for qualitative and quantitative characterization of complex mixtures using the resolving power of chromatography and the ability of mass spectrometry to identify separated compounds.

The main fields of application of UPLC are chemistry, pharmacy, foodstuffs, biochemistry, and the chemistry of compounds used in the heavy metal industry [8–12]. The UPLC systems are also important tools in research and production. For example, they are used to detect the presence of performance-enhancing drugs in samples provided by athletes [13] to check the purity of manufactured drugs [14] or in the food industry to determine the concentration of important ingredients (e.g., vitamins in juices) [15]. These methods can be used to assess the number of ingredients present in a sample as well as to determine

purity in the process of ensuring the quality control of test compounds [16]. For example, many dishonest spice producers use Sudan as a red dye to improve the aesthetic value of their products. The existing UPLC method for identifying this dye in food products can give a quick and truthful answer [17]. UPLC is also used to separate and identify amino acids, nucleic acids, proteins, hydrocarbons, pesticides, carbohydrates, antibiotics, steroids, and many other compounds [18]. UPLC apparatuses also prove themselves during the determination of additives used in electroplating [19] and the analysis of explosives [20]. In the field of ecology, the UPLC–MS method is known to determine the level of pesticides in groundwater [21] as well as to analyze wastewater in terms of the content of medicinal substances [22].

The UPLC method finds more and more applications in the field of drug substance analysis, especially drug identification. Many researchers attempt to modify and transfer the assay conditions from the commonly used HPLC method to the UPLC method [23]. When analyzing the available publications, it can be noticed that UPLC systems are starting to displace standard HPLC systems, especially in the pharmaceutical industry [24]. Thanks to the increased resolution, new analytical procedures are refined, in many cases based on existing methods, eliminating the need for re-analysis.

Our goal was to collect the latest applications of the UPLC analytical method in drug quality analysis that appeared after 2000. We focused on drugs commonly used in medicine that belong to various pharmacological groups. Rational prescribing based on clinical pharmacology is essential if the right drug is to be administered to the right patient at the right time. This requires, *inter alia*, specific knowledge about the drugs used, especially their quality, which is directly related to the safety of use. This manuscript shows that the UPLC technique is now an increasingly used tool for assessing the quality of drugs and determining the identity and content of active substances. It also allows the monitoring of substances and finished products during their processing and storage. The collected information is a summary of the available analytical procedures using the UPLC technique in the analysis of biologically active compounds belonging to various therapeutic groups in pharmaceutical preparations.

## 2. Conditions for UPLC Analysis of Medicinal Substances

### 2.1. Cardiovascular Drugs

Cardiovascular drugs are substances used in diseases related to the structure and function of the heart and blood vessels, such as arrhythmias, blood clots, coronary artery disease, high or low blood pressure, high cholesterol, heart failure, stroke, circulatory disorders, and others. These include a large number of prescription drugs, and the type of cardiovascular disease the patient has will determine which drug to use [25–28]. Some examples of drugs most commonly used in cardiovascular medicine include anticoagulants (e.g., heparin, warfarin, etc.), antiplatelet drugs (e.g., clopidogrel and lopidogrel), angiotensin converting enzyme (ACE) inhibitors (e.g., captopril and enalapril), angiotensin receptor blockers (ARBs and sartan) such as candesartan or valsartan, beta-blockers (e.g., bisoprolol and sotalol), calcium channel blockers (e.g., amlodipine and diltiazem), diuretics (e.g., chlorothiazide and furosemide), vasodilators such as isosorbide and hydralazine, digoxin used to treat arrhythmias to slow the heart rate, and other drugs used to regulate abnormal heart rhythms that include, but are not limited to, quinidine, lidocaine, amiodarone, and adenosine. Table 1 presents details of the analysis of drugs from this therapeutic group for which the UPLC technique was used.

**Table 1.** UPLC technique in the analysis of cardiovascular drugs.

| Active Substance | Sample | Column | Mobile Phase (Gradient: Time [min]/%B) | Flow Rate | Detection | Comments | Ref |
|---|---|---|---|---|---|---|---|
| Valsartan Hydrochlorothiazyd | combined tablets | Kromasil Eternity C-18 (50 × 2.1 mm, 3.5 μm) | A-methanol; B-0.1% triethylamine pH3; A:B (75:25, $v/v$) | 0.6 mL/min | UV 225 nm | assay | [29] |
| Lodenafil | tablets | BEH C18 (50 × 2.1 mm, 1.7 μm) | A-methanol; B-0.1% formic acid pH4; A:B (55:45, $v/v$) | 0.4 mL/min | MS | photodegradtion; cytotoxicity; determination of degradation products | [30] |
| Ezetynibe Simvastatin | tablets | Kromasil Eternity TM C18 (50 × 2.1 mm, 2.5 μm) | A-acetonitrile; B-0.01 M ammonium acetate buffer pH6.7; Gradient elution | 0.35 mL/min | UV 235 nm | degradation study | [31] |
| Trandolapril | substance | BEH C18 (100 × 2.1 mm, 1.7 μm) | A-ammonium bicarbonate in water B-acetonitrile; A:B (68:32, $v/v$) | 0.4 mL/min | UV 211 nm; QTOF-MS | degradation study | [32] |
| Pitawastatin | substance | BEH C18 (100×2.1 mm, 1.7 μm) | A-phosphate buffer; B-acetonitrile; Gradient: 0/45, 2/45, 2.5/100, 4/100, 4.5/45, 5/45 | 0.3 mL/min | UV 245 nm | degradation study | [33] |
| Valsartan | tablets, substance | BEH C18 (100×2.1 mm, 1.7 μm) | A-1% acetic acid buffer, acetonitrile (90:10, $v/v$) B-acetic acid buffer, acetonitrile (10:90, $v/v$) Gradient: 0.01/20, 1/40, 3.5/55, 6.5/80, 8.5/80, 8.9/20, 9.5/20 | 0.3 mL/min | UV 225 nm | degradation study | [34] |
| Amlodipine Benazepril | combined tablets | BEH C8 (100 × 2.1 mm, 1.7 μm) | A-phosphate buffer pH3 B-acetonitrile, methanol (1:1, $v/v$); A:B (45:55, $v/v$) | 0.3 mL/min | UV 237 nm | different columns tests | [35] |
| Atorvastatin | tablets, substance | Zorbax Extended C18 (50 × 3.0 mm, 1.8 μm) | A-acetonitrile; B-phosphoric acid Gradient: 0.01/50, 8/90, 10.1/10 | 0.5 mL/min | UV | assay | [36] |
| Fosinopril | substance | HSS C18 (100 × 2.1 mm, 1.8 μm) | A-phosphate buffer; B-acetonitrile; Gradient: 0.01/20, 12/80, 20/80, 20.2/20, 25/20 | 0.1 mL/min | UV 205 nm | monitoring during production; degradation study; detection of impurities | [37] |
| Olmesartan Amlodypine Hydrochlortiazide | tablets | Zorbax SB Phenyl (50 × 2.1 mm, 1.8 μm) | A-0.053 M sodium perchlorate, acetonitrile (90:10, $v/v$) B-0.053 M sodium perchlorate acetonitrile (10:90, $v/v$) Gradient: 0/10, 2/50, 4/80, 6/10 | 0.7 mL/min | UV 271, 215, 237 nm | combined tablet; degradation study | [38] |
| Atorvastatin Fenofibrate | combined tablets | BEH C18 (100 × 2.1 mm, 1.7 μm) | A-acetate buffer; B-acetonitrile; Gradient: 0/50, 1/70, 1.4/85, 2.2/50 | 0.5 mL/min | UV 247 nm | detection of impurities | [39] |
| Bisoprolol Hydrochlortiazide | combined tablets, urine | BEH C18 (50 × 2.1 mm, 1.7 μm) | A-acetonitrile; B-phosphoric buffer Gradient: 0/85, 0.6/80, 1.4/40 | 0.7 or 0.9 mL/min | UV 225 nm | assay | [40] |
| Amlodipine Atorvastatin | tablets | Kromasil C18, (50 × 2.1 mm, 3.5 μm) | A-acetonitrile; B-triethylamine Gradient: 0/30, 0.5/36, 1.3/60, 2.05/30 | 0.8 mL/min | UV 240 nm | degradation study | [41] |
| Telmisartan Amlodipine Hydrochlorotiazide | tablets | BEH C18 (100 × 2.1 mm, 1.7 μm) | A-0.053 M sodium perchlorate, acetonitrile (90:10, $v/v$) B-0.053 M sodium perchlorate, acetonitrile (20:80, $v/v$) Gradient: 0/5, 1.2/5, 1.6/40, 4/40, 4.1/5, 4.5/5 | 0.6 mL/min | UV 237, 271 nm | assay | [42] |
| Moxonidine | tablets | C18 Hypersil Gold (100 × 2.1 mm, 1.9 μm) | A-methanol; B-ammonium acetate buffer (10 mM, pH3.43); A:B (0.9:99.1, $v/v$) or (6:94, $v/v$) | 0.87 mL/min | UV 255 nm; MS | degradation study | [43] |
| Simvastatin | tablets | BEH C18 (100 × 2.1 mm, 1.7 μm) | A-acetonitrile; B-ammonium acetate Gradient: 0–5/50–0, 5.5/0, 5.6/50 | 0.8 mL/min | MS | assay; differences in product series | [44] |
| Ticlopidine | tablets | Zorbax SB-C18 (50 × 4.6 mm, 1.8 μm) | A-methanol; B-0.01 M ammonium acetate buffer pH5; A:B (80:20, $v/v$) | 0.8 mL/min | UV 235 nm | degradation study | [45] |
| Telmisartan | substance | BEH C18 (150 × 2.1 mm, 1.7 μm) | A-acetonitrile; B-water; A:B (70:30, $v/v$) | 0.2 mL/min | UV 230 nm | degradation study | [46] |
| Metoprolol Atorvastatin Ramipril | combined tablets | Zorbax XDB-C18 (50 × 4.6 mm, 1.8 μm) | A-0.0045 M sodium lauryl sulfate; B-acetonitrile A:B (50:50 $v/v$) | 1.0 mL/min | UV 210 nm | assay | [47] |
| Rosuvastatin | tablets | BEH C18 (100 × 2.1 mm, 1.7 μm) | A-0.1% trifluoroacetic acid; B-acetonitrile Gradient: 0/55, 3.5/60, 6.5/85, 7.5/85, 7.6/55, 10/55 | 0.3 mL/min | UV 240 nm | degradation study; identification of degradation products | [48] |
| Bisoprolol Amlodipine | substance | B CSH C18 (50 × 2.1 mm, 1.7 μm) | A-phosphate buffer; B-acetonitrile Gradient: 0–10/10–90 | 0.5 mL/min | UV | computer simulation | [49] |
| Rivaroxaban | tablets | Eclipse Plus C18 (2.1 × 50 mm, 1.8 μm) | A-water adjusted to pH4 with ammonium hydroxide B-acetonitryl; A:B (63:37 $v/v$) | 0.2 mL/min | QTOF-MS | degradation study; identification of degradation products | [50] |

**Table 1.** *Cont.*

| Active Substance | Sample | Column | Mobile Phase (Gradient: Time [min]/%B) | Flow Rate | Detection | Comments | Ref |
|---|---|---|---|---|---|---|---|
| Telmisartan | substance | BEH C18 (100 × 2.1 mm, 1.7 μm) | A-potassium phosphate<br>B-acetonitrile, methanol, water (7.5:1.5:1.0) *v/v/v*<br>Gradient: 0/55, 4/55, 5/70, 7.5/70, 7.7/55, 8/55 | 0.33 mL/min | UV 235 nm | degradation study; analysis of impurities | [51] |
| Perindoprill | tablets | Poroshell 120 Hilic (4 × 150 mm, 2.7 μm) | A-acetonitrile; B-0.1% formic acid; A:B (20:80 *v/v*) | 1.0 mL/min | UV 230 nm | Separation of cis and trans isomers; degradation study | [52] |
| Enalapril Hydrochlorotiazide | tablets | BEH C18 (100 × 2.1 mm, 1.7 μm) | A-phosphoric acid; B-acetonitrile<br>Gradient: 0/5, 2/20, 4/60, 5/60, 6/5 | 0.5 mL/min | UV 210 nm | degradation study | [53] |
| Oxprenolol Metoprolol Acebutolol Atenolol Propranolol Pindolol Alprenolol | substance | BEH C18 (100 × 2.1 mm, 1.7 μm) | A-0.1% trifluoroacetic acid in water<br>B-0.1% trifluoroacetic acid in acetonitryl<br>Gradient: 0–10/20–50 | 0.5 mL/min | UV 270 nm;<br>MS;<br>NMR | comparison of various detectors | [54] |
| Perindopril Indapamide | tablets | BEH C18 (50 × 2.1 mm, 1.7 μm) | A-0.01% formic acid in water pH4<br>B-acetic acid, acetonitrile (40:60 *v/v*); Gradient: 0.01/15, 2.5/30, 7/30, 9/70, 10/70, 11/15, 13/15 | 0.3 mL/min | UV 227 nm | degradation study | [55] |
| Rivaroxaban Enalapril | plasma | BEH C18 (50 × 2.1 mm, 1.7 μm) | A-acetonitrile; B-0.1% formic acid<br>Gradient: 0–0.5/80–5, 0.5–2.9/5 2.9–3/5–80, 3–4/80 | 0.3 mL/min | MS | pharmacokinetics study; interactions | [56] |
| Atorvastatin Acetylosalicylic acid Clopidogrel | combined capsules | Eclipse plus C18 (100 × 2.1 mm, 1.7 μm) | A-20 mM anhydrous KH2PO4 buffer containing 0.2% triethylamine pH2.7 with o-phosphoric acid<br>B-acetonitrile; A:B (55:45, *v/v*) | 0.3 mL/min | DAD 240, 220 nm | Comparison with HPLC; analysis of impurities | [57] |
| Azilsartan | tablets | BEH C18 (100 × 2.1 mm, 1.7 μm) | A-0.1% o-phosphoric acid in water pH3B-acetonitrile; Gradient: 0/35, 5/60, 7/60, 7.1/35, 10/35 | 0.5 mL/min | UV 215 nm | assay | [58] |
| Amlodipine Olmesartan | combined tablets | BEH C8 (100 × 2.1 mm, 1.7 μm) | A-0.1% orthophosphoric acid in water; B-acetonitrile<br>Gradient: 0/22, 6/35, 10/60, 11.5/70, 12/70, 12.5/22, 15/22 | 0.5–0.7 mL/min | UV 237 nm | degradation study; analysis of impurities | [59] |
| Dabigatran | capsules | HSS-T3 (100 × 2.1 mm, 1.8 μm) | A-0.1% orthophosphoric acid in water pH3.5 with triethyl amine; B-acetonitril<br>Gradient: 0/20, 12/60, 12.1/60, 15/60, 15.1/20, 18/20 | 0.18 mL/min | UV 290 nm | degradation study; analysis of impurities | [60] |
| Perindopril Amlodipine | combined tablets | Agilent SD C18 (50 × 3.0 mm, 1.8 μm) | A-0.1% perchloric acid; B-acetonitrile<br>Gradient: 0.01/15, 2.5/30, 6/34, 8.5/60, 12/90, 12.5/90, 13/15, 15/15 | 0.8 mL/min | UV 215 nm | degradation study; analysis of impurities | [61] |
| Perindopril Indapamide | tablets | Agilent SB 18 (50 × 3.0 mm, 1.5 μm) | A-0.1% perchloric acid; B-acetonitrile<br>Gradient: 0.01/15, 2.5/30, 7/30, 9/70, 10/70, 11/15, 13/15 | 0.8 mL/min | UV 215 nm | degradation study; analysis of impurities | [62] |
| Indapamide | substance | Acquity HSS T3 (100 × 2.1 mm, 1.8 μm) | A-water with 0.1% formic acid<br>B-acetonitrile with 0.1% formic acid<br>Gradient: 0/10, 2/10, 8/50, 9/50, 10/80, 11/80, 12/10, 15/10 | 0.5 mL/min | UV 274 nm;<br>MS | degradation study | [63] |
| Lenvatinib Telmisartan | substance, plasma | X Select HSS T3 (100 × 2.1 mm, 2.5 μm) | A-water with 0.1% formic acid and 5 mM ammonium acetate; B-acetonitrile with 0.1% formic acid<br>Gradient: 2/60, 2–3/60–90, 3–4/90, 4–4.1/910–60, 4.1–5.1/60 | 0.25 mL/min | MS-MS | assay | [64] |

### 2.2. Nonsteroidal Anti-Inflammatory Drugs (NSAIDs)

NSAIDs are widely used around the world due to their wide availability and range of effects. They are usually given to control pain, fever, and inflammation. They are often used to relieve the symptoms of headaches, toothaches, painful periods, sprains, colds and flu, arthritis, and other causes of long-term pain [65–68]. There are many different NSAIDs available, but they all work in the same way by blocking cyclooxygenase (COX) enzymes, which are responsible for the production of prostaglandins, a group of compounds that control many different processes in the body. NSAIDs are a group of compounds with heterogeneous chemical structures and applications. However, they all have at least three things in common: identical pharmacological properties, the same basic mechanism of action, and similar side effects. According to chemical structure, NSAIDs can be classified into salicylates (e.g., acetylsalicylic acid), indole acetic acid derivatives (e.g., indomethacin), phenylacetic acid derivatives (e.g., diclofenac), phenylpropionic acid derivatives (e.g., naproxen), fenamic acid derivatives (e.g., mefenamic acid), enolic acid derivatives (e.g., piroxicam), and others. Another classification (important for clinicians) based on the ability to inhibit COX distinguishes non-selective COX-1 inhibitors (e.g., ibuprofen, diclofenac, naproxen, and indomethacin), selective COX-1 inhibitors (such as acetylsalicylic acid at cardiac doses), selective COX-2 inhibitors (coxibs), and preferential COX-2 inhibitors (e.g., meloxicam, nimesulide). All NSAIDs are usually used to treat pain, fever, and inflammation. Ibuprofen, nabumetone, coxibs, and diclofenac are most commonly used in rheumatology, indomethacin in neonatology, celecoxib for familial adenomatous polyposis, and ketorolac for acute pain that usually requires narcotics. Aspirin is a unique NSAID, not only because of its many uses but because it is the only NSAID that inhibits the clotting of blood for a prolonged period of time. The parameters of the UPLC method used to analyze these drugs are summarized in Table 2.

**Table 2.** UPLC technique in the analysis of NSAIDs and antibiotics.

| Active Substance | Sample | Column | Mobile Phase (Gradient: Time [min]/%B) | Flow Rate | Detection | Comments | Ref |
|---|---|---|---|---|---|---|---|
| Diclofenac | gel, substance | BEH C18 (50 × 2.1 mm, 1.7 μm) BEH C18 (100 × 2.1 mm, 1.7 μm) | A-methanol; B-phosphoric acid pH2.5 A:B (65:35, *v/v*) | 0.4 or 0.45 mL/min | UV 254 nm | pollutants study; comparison of various columns | [69] |
| Ibuprofen Diphenhydramine | combined tablets | BEH C18 (50 × 2.1 mm, 1.7 μm) | A-0.1% triethylamine buffer pH3.2 with phosphoric acid, acetonitrile (80:20, *v/v*) B-0.1% triethylamine buffer pH3.2 with phosphoric acid, acetonitrile (50:50 *v/v*) Gradient: 0/0, 7.5/50, 17/50, 17.5/0, 20/0 | 0.4 mL/min | UV 220 nm | degradation study | [70] |
| Nabumeton | tablets | BEH C18 (100 × 2.1 mm, 1.7 μm) | A-5 mM ammonium acetate B-acetonitrile; A:B (25:75, *v/v*) | 0.3 mL/min | UV 230 nm | assay | [71] |
| Ketoprofen | microdialyzate, human skin | BEH C18 (100 × 2.1 mm, 1.7 μm) | A-acetonitrile; B-methanol; C-water A:B:C (60:20:20, *v/v/v*) | 0.3 mL/min | UV 255 nm; MS | assay (very high sensitivity) | [72] |
| Naproxen | tablets | BEH C18 (50 × 4.6 mm, 1.7 μm) | A-dihydrophosphate buffer, methanol (90:10, *v/v*); B-methanol, acetonitryl (50:50, *v/v*) Gradient: 0.01/20, 2/30, 5/50, 6/70, 8.5/70, 9.5/20, 11/20 | 0.3 mL/min | UV 260 nm | degradation study | [73] |
| Levofloxacin | tablets | BEH C18 (100 × 2.1 mm, 1.7 μm) | A-buffer (20 mM KH$_2$PO$_4$ + 1 mL triethylamine in 1 L of water pH2.5 with orthophosphoric acid B-acetonitrile; A:B (77:23 *v/v*) | 0.4 mL/min | UV 294 nm | degradation study | [74] |
| Sparfloxacin | substance, tablets, eye drops | HSS T-3 (100 × 2.1 mm, 1.8 μm) | A-orthophosphoric acid; B-water Gradient: 1/10, 2/10, 3/25, 4/10, 5/10 | 0.5 mL/min | UV 290 nm | assay | [75] |
| Isoniazid Pirazynamide Rifampicin | combined tablets | Shield RP18 (50 × 2.1 mm, 1.7 μm) | A-50 mM phosphate buffer; B-acetonitrile Gradient: 0–0.3/2, 0.3–1/2–40, 1–1.2/40, 1.2–1.7/40–2 | 1.0 or 1.5 mL/min | UV 254 nm | assay; column testing at different temperatures | [76] |
| Moxifloxacine | tablets | HSS C-18 (100 × 2.1 mm, 1.8 μm) | A-phosphate buffer; B-methanol; C-acetonitrile A:B:C (60:20:20, *v/v/v*) | 0.3 mL/min | UV 296 nm | degradation study | [77] |
| Doripenem Meropenem Tebipenem | substance | Kinetex C18 (100 × 2.1 mm, 1.7, 2.6, 5 μm) | A-acetonitrile; B-ammonium acetate A:B (4:96 or 10:90 or 7:93, *v/v*) | 0.5 or 1.0 mL/min | UV 298 nm | degradation study | [78] |
| Cefuroxim | tablets | Kinetex C-18 (100 × 2.1 mm, 1.7 μm) | A-0.1% formic acid; B-methanol A:B (88:12, *v/v*) | 0.7 mL/min | UV 278 nm; MS | determination of diastereomers in crystalline, amorphous and tablet form; degradation study | [79] |
| Ceftalozone Tazobactam | plasma | BEH-Shield RP18 (100 × 2.1 mm, 1.7 μm) | A-0.1% formic acid in water B-0.1% formic acid in acetonitrile Gradient: 0–0.5/2, 0.5–2/2–50, 2–2.5/50–98 | 0.4 mL/min | MS TQD | assay | [80] |
| Amoxicillin Clavulanate | tablets | ACQUITY BEH C18 (50 × 2.1 mm, 1.7 μm) | A-buffer solution pH4.4; B-methanol A:B (98:2, *v/v*) | 0.1 mL/min | UV 220 nm | comparison with HPLC | [81] |
| Acetaminophen Tramadol | tablets | HSS T3 (100 × 2.1 mm, 1.8 μm) | A-0.1% perchloric acid in water; B-acetonitrile Gradient: 0/10, 4/10, 8/15, 15/25, 25/35, 25.1/10 | 0.5 mL/min | UV 215 nm | degradation study, analysis of impurities | [82] |
| Diclofenac Paracetamol Camylofin | combined tablets | HSS C18 (50 × 2.1 mm, 1.8 μm) | A-20 mM ammonium acetate buffer pH3 B-methanol; A:B (33:67, *v/v*) | 0.25 mL/min | UV 220 nm | degradation study; transferred from HPLC | [83] |
| Paracetamol Ibuprofen | combined tablets | BEH C18 (100 × 2.1 mm, 1.7 μm) | A-0.01% aqueous triethylamine pH7 B-methanol Gradient: 0–2.5/2, 2.5–4.5/2–50, 4.5–7/50–98 | 0.2 mL/min | UV DAD230 nm | comparison with HPLC; analysis of impurities | [84] |
| Naproxen | gelatin capsules | BEH C18 (100 × 2.1 mm, 1.7 μm) | A-0.1% orthophosphoric acid in water pH3 B-acetonitrile Gradient: 0/35, 3/35, 10/70, 10.5/35, 13/35 | 0.5 mL/min | UV 230 nm | degradation study; analysis of impurities | [85] |
| Ibuprofen | human plasma | BEH Phenyl (150 × 2.1 mm, 1.7 μm) | A-10 mM ammonium acetate with 0.1% formic acid in water B-10 mM ammonium acetate with 0.1% formic acid in acetonitrile, methanol (64:36, *v/v*) Gradient: 0–12/65, 12.1–14/65–100 | 0.2–0.5 mL/min | MS/MS | degradation study; transferred from HPLC | [86] |
| Cefuroxim | injections | Shim-pack XR-ODS (75 × 3 mm, 2.2 μm) | A-acetonitrile; B-formic acid A:B (70:30, *v/v*) | 0.3 mL/min | MS/MS | analysis of impurities | [79] |
| Ibuprofen Pseudoephedrine Chlorpheniramine | tablet | Acquity BEH (50 × 2.1 mm, 1.7 μm) | A-0.1% formic acid in water B-0.1% formic acid in methanol Gradient: 1/5, 2/5–80, 1/80 | 0.3 mL/min | MS | assay | [87] |
| Amoxicillin | tablet | BEH C18 (100 × 2.1 mm, 1.7 μm) | A-phosphate buffer pH5; B-methanol A:B (95:5, *v/v*) | 0.3 mL/min | UV 230 nm | assay | [88] |
| Ibuprofen | substance | Accucore XL C18 (150 × 4.6 mm, 4 μm) | A-water with 1% chloroacetic acid pH3 B-acetonitrile; A:B (40:60, *v/v*) | 2.0 mL/min | UV 254 nm | determination of impurities | [89] |
| Antibiotics [1] | substance, plasma | Acquity HSS T3 (50 × 2.1 mm, 1.8 μm) | A-water with 0.1% formic acid B-acetonitrile with 0.1% formic acid Gradient: 0/0, 3.6/85.5, 3.601/95, 4.1/95, 4.11–5.5/0 | 0.3 mL/min | MS-MS | assay | [90] |
| Ibuprofen Famotidine | tablet | Acquity BEH C-18 (50 × 2.1 mm, 1.7 μm) | A-50 mM sodium acetate buffer pH5.5 B-methanol; A:B (25:75, *v/v*) | 0.3 mL/min | UV 260 nm | assay | [91] |
| Lansoprazole Naproxen | substance, tablet | Phenomenex Luna C18 (250 × 4.6 mm, 5 μm) | A-methanol; B-water; A:B (8:2, *v/v*) | 1.0 mL/min | PDA | assay | [92] |
| NSAIDs [2] | preparations | Hypersil Golden C18 | A-5 mM ammonium formate B-methanol; Gradient | 0.2 mL/min | MS-MS | assay | [93] |
| Diclofenac | substance, tablet | Acquity BEH C18 (50 × 2.5 mm, 1.7 μm) | A-0.05 M acetate buffer pH2.5 B-acetonitrile; A:B (50:50, *v/v*) | 0.5 mL/min | PDA 254 nm | degradation study | [94] |

[1] Amoxicillin, Aztreonam, Cefazolin, Cefepime, Cefotaxime, Cefoxotin, Ceftazidine, Ciprofloxacin, Clindamycin, Dapomycin, Ertapenem, Linezolid, Meropenem, Ofloxacin, Piperacillin. [2] Acetaminophen, Acetylsalicylic acid, Aminopyrine, Meloxicam, Ibuprofen, Naproxen, Nimesulide, Diclofenac, Indomethacin, Ketoprofen, Celecoxib.

### 2.3. Antibiotics

Antibiotics are one of the most commonly used drug classes to treat bacterial infections. They work by destroying or slowing down the growth of bacteria [95–98]. A class of antibiotics is a group of different drug substances with similar chemical and pharmacological properties. Their chemical structures may look similar, and drugs of the same class may kill the same or related bacteria. The main classes of antibiotics are penicillins, including five classes, such as aminopenicillins, pseudomone penicillins, beta-lactamase inhibitors, natural penicillins, and penicillinase-resistant penicillins (e.g., amoxicillin, ampicillin, etc.); tetracyclines with a broad spectrum of activity against many bacteria (among others such as doxycycline and tetracycline); cephalosporins (e.g., cefaclor and ceftriaxone); quinolones (fluoroquinolones; e.g., ciprofloxacin and moxifloxacin); lincomycins (e.g., clindamycin and lincomycin); macrolides used as an alternative for people allergic to penicillin (e.g., clarithromycin and erythromycin); sulfonamidessuch as sulfamethoxazole and trimethoprim; glycopeptides used to treat methicillin-resistant *Staphylococcus aureus* (MRSA) (e.g., dalbavancin and vancomycin); aminoglycosides (among others such as gentamicin and tobramycin); and carbapenems often used as "last-line" measures to prevent resistance (e.g., imipenem and ertapenem). Details on the conditions for the analysis of antibiotics by the UPLC method are presented in Table 2.

### 2.4. Antifungal and Anthelmintic Drugs

Most antifungal drugs interfere with the biosynthesis or integrity of ergosterol, the major sterol in the fungal cell membrane. Others disrupt the fungal cell wall. Based on their mechanism of action, they can be classified into five classes: polyenes, azoles, allylamines, echinocandin, and other agents (including griseofulvin and flucytosine) [99–101].

Polyene antifungal drugs interact with sterols in the cell membrane (for example, amphotericin B, nystatin, or pimaricin). Azoles are the most widely used antifungal drugs and act mainly by inhibiting 14α-demethylase, the fungus cytochrome P450 enzyme. There are two groups in clinical use: imidazoles (ketoconazole, miconazole, and clotrimazole) and triazoles (fluconazole, itraconazole, and voriconazole). Newer antifungal drugs include the echinocandin class (e.g., caspofungin) and second generation triazoles (e.g., voriconazole and posaconazole). Allylamines (naphtifine and terbinafine) inhibit ergosterol biosynthesis at the level of squalene epoxidase. The drug morpholine, amorolfine, inhibits the same pathway at a later stage. Griseofulvin is an antifungal antibiotic produced by *Penicillium griseofulvum*, active in vitro against most dermatophytes. Anthelmintics are a type of medicine used to treat helminth infections in animals. The main drugs used in the treatment of tapeworm infections are albendazole and praziquantel. Other drugs in this group include quinacrine, diethylcarbamazine, mebendazole, or phenothiazine. An antibiotic, hygromycin, is also used as an anthelmintic agent in the form of a livestock feed additive.

Miconazole (as a substance) was determined using the column Thermo Scientific Hypersil Gold C18 (50 × 4.6 mm, 1.9 μm) as the stationary phase, and isocratic elution of the mobile phase, containing solvents acetonitrile, methanol, and ammonium acetate (30:32:38, *v/v*). The separation was carried out with a flow rate of 2.5 mL/min, and spectrophotometric detection was carried out at a wavelength of 235 nm [102]. Dongre et al. compared the condition of UPLC assays with the HPLC technique based on the determination of the primaquine substance. Analysis was carried out with a BAH C18 (50 × 2.1 mm, 1.7 μm) column, and a mixture of 0.01% aqueous trifluroacetic acid and acetonitrile (75:25, *v/v*) as a mobile phase, with a flow rate of 0.5 mL/min. Detection was in the UV range at 265 nm [103]. The mixture of the nine active substances (flubendazole, pipamperone, cinnarizine, ketoconazole, miconazole, rabeprazole, itraconazole, domperidone, and propiconazole) was analyzed in surface waters using HSS T3 (100 × 2.1 mm, 1.8 μm) column, and gradient elution of the mobile phase (A-water:acetonitrile (95:5, *v/v*); B-water:acetonitrile (5:95, *v/v*)) with a flow rate of 0.5 mL/min. Gradient conditions were as follows: 0–4.38 min, linear from 20 to 100% B; 4.38–6.46 min, isocratic 100% B; 6.46–6.67 min, linear from 100 to 20% B; 6.67–9.59 min, isocratic 20% B. The authors noted the matrix effect seen during

HPLC analyzes. In the case of the UPLC technique with MS detection, using the internal standard, the matrix effect does not occur, which greatly simplifies the procedure. They wanted to limit the matrix effect in quantitative UPLC–MS determinations which is very evident in HPLC [104]. Whereas secnidazole, fluconazole, and azithromycin (in the form of tablets) were determined using a BEH-Shield RP18 (100 × 2.1 mm, 1.7 μm) column. The mobile phase containing a phosphate buffer (A) and acetonitrile (B) with linear gradient eluent program (time [min]/%B: 0/5, 1.5/5, 3/30, 5/90, 8/90, 9/5, 10/5) was used with a flow rate of 0.3 mL/min and UV detection at 210 nm. The authors also analyzed the drug degradation process, finding the degradation in an alkaline environment [105]. Elkady et al. developed a method for the determination of tinidazole and hydrocortisone in substances, vaginal tablets, and cream. The separation of components was carried out on an Acquity Eclipse plus C18 (100 × 2.1 mm, 1.7 μm) column using a mobile phase with the following composition: 0.02 M anhydrous $KH_2PO_4$ (with 0.2% triethylamine) pH6 with orthophosphoric acid (A) and acetonitrile (B) and flow rate 0.3 mL/min. The eluent gradient program was as follows: 0/50, 2/70, 5.6/70, 5.7/50, 7/50 (time [min]/%B). The established conditions and spectrophotometric detection in UV at a wavelength of 220 nm also allowed for the analysis of impurities present in the tested material [106]. Clotrimazole in substance and human plasma was analyzed on an Acquity BEH C-18 (50 × 2.1 mm, 1.7 μm) column and a mixture of water with 0.2% ammonium acetate and 0.1% formic acid (A) and methanol (B) in a volume ratio 18:82 (A:B). The flow rate of the eluent was 0.1 and 0.7 mL/min, and MS-MS detection was performed [107].

### 2.5. Antipsychotics, Antidepressants, and Drugs Used in Diseases of the Nervous System

Neuropsychiatric symptoms are often associated with cognitive decline. Antipsychotics are a type of medication that is available with a prescription to treat certain types of mental health problems, such as schizophrenia, schizoaffective disorder, certain forms of bipolar disorder, depression, psychotic symptoms of personality disorder, and Alzheimer's disease. Some antipsychotics are also used to treat other health problems, including physical problems (e.g., persistent hiccups, problems with balance, and nausea), agitation, and psychotic experiences in dementia. Antipsychotic drugs can help calm and control symptoms but do not treat the underlying disease [108–111]. When overused for a long period of time, they can have serious side effects. They are divided into two main groups: typical (first-generation) and atypical (second-generation). The main difference between them is that atypical drugs block dopamine, and typical drugs block dopamine and affect serotonin levels. Atypical antipsychotics, usually the drugs of first choice for the treatment of schizophrenia, include risperidone, quetiapine, ziprasidone, aripiprazole, and clozapine. Typical antipsychotics are older-generation substances such as chlorpromazine, flupentixol, haloperidol, or loxapine.

Antidepressants help reduce the symptoms of depressive disorders by changing the chemical balance of neurotransmitters in the brain [112–114]. The change in mood and behavior is due to a chemical imbalance. Neurotransmitters (i.e., serotonin, dopamine, and norepinephrine) are the link between neurons. Antidepressants inhibit the reuptake of neurotransmitters by selective receptors, thus increasing the concentration of a specific neurotransmitter around the nerves. They are used not only in the treatment of depression but also nervousness, diabetic peripheral neuropathic pain, post-traumatic stress disorder, etc. Antidepressant drugs can be divided into five groups: tricyclic antidepressants (TCAs), selective serotonin reuptake inhibitors (SSRIs), monoamine oxidase inhibitors (MAOIs), serotonin norepinephrine reuptake inhibitors (SNRIs), and norepinephrine and specific serotonergic antidepressants (NASSA). They include bupropion, clomipramine, amitriptyline, fluoxetine, doxepin, desipramine, and moclobemide.

The nervous system is a complex system that coordinates the activities of the entire body. Clinical neuroscience is the part of medicine that focuses on the nervous system (central and peripheral) [115–117]. This system can be affected by many different conditions, for example, benign and malignant neoplasms, degenerative diseases (e.g., Alzheimer's

and Parkinson's disease) or pituitary disorders, epilepsy, and demyelinating diseases (e.g., multiple sclerosis). Currently available treatments for many diseases of the nervous system focus primarily on relieving symptoms. The symptoms of Parkinson's disease are often treated with co-beneldopa, co-kareldopa, or ropinirole. Alzheimer's disease progression can be slowed down by donepezil or memantine. Seizures can be controlled with anticonvulsants such as carbamazepine or levetiracetam. The conditions of UPLC analysis of drugs from the above-mentioned groups are summarized in Table 3.

**Table 3.** UPLC technique in the analysis of antipsychotics, antidepressants, and drugs used in diseases of the nervous system.

| Active Substance | Sample | Column | Mobile Phase (Gradient: Time [min]/%B) | Flow Rate | Detection | Comments | Ref |
|---|---|---|---|---|---|---|---|
| Paliperidon | tablets | BEH C18 (100 × 2.1 mm, 1.7 μm) | A-phosphate buffer; B-acetonitrile, water (9:1, $v/v$) Gradient: 0.01/16, 6/16 | 0.45 mL/min | UV 238 nm | degradation study | [118] |
| Venlafaxine | capsules | BEH C18 (100 × 2.1 mm, 2.0 μm) | A-dipotassium hydrogen phosphate B-acetonitrile; A:B (30:70, $v/v$) | 0.75 mL/min | UV 227 nm | assay | [119] |
| Olanzapine | tablets, substance | BEH C18 (100 × 2.1 mm, 1.7 μm) | A-triethylamine buffer pH6.8, acetonitrile, methanol (50:20:30, $v/v/v$); B-water, acetonitrile (10:90, $v/v$) Gradient: 0.01/0, 5/20, 6.5/90, 8/100, 9/0, 10/0 | 0.3 mL/min | UV 250 nm | degradation study; analysis of impurities; comparison with HPLC | [120] |
| Quetiapine Aripiprazole Perospirone | substance | BEH C18 (100 × 2.1 mm, 1.7 μm) | A-acetonitrile; B-ammonium acetate A:B (62:38, $v/v$) | 0.3 mL/min | MS | assay | [121] |
| Duloxetin | tablets | Zorbax XDB C-18 (50 × 4.6 mm, 1.8 μm) | A-0.01 M $KH_2PO_4$ buffer pH4, tetrahydrofuran, methanol (67:23:10, $v/v/v$) B-0.01 M $KH_2PO_4$ buffer pH4, acetonitrile (60:40 $v/v$) Gradient: 0/0, 6/0, 8/100, 13/100, 14/0, 16/0 | 0.6 mL/min | UV 236 nm | degradation study | [122] |
| Quetiapine | tablets | Agilent Eclipse Plus C18 (50 × 2.1 mm, 1.8 μm) | A-triethylamine in water pH7.2 B-acetonitrile, methanol (80:20, $v/v$) Gradient: 0/30, 0.5/30, 3/95, 4/95, 4.1/30, 5/30 | 0.5 mL/min | UV 252 nm | determination of impurities | [123] |
| Aripiprazole | tablets | BEH C8 (50 × 2.1 mm, 1.7 μm) | A-acetonitrile; B-20 mM ammonium acetate A:B (90:10, $v/v$) | 0.25 mL/min | UV 240 nm | comparison with HPLC | [124] |
| Quetiapine | plasma | BEH Phenyl (50 × 2.1mm, 1.7 μm) | A-10 mM ammonium acetate with 0.3% formic acid in water; B-acetonitrile; A:B (70:30, $v/v$) | 0.5 mL/min | MS | bioequivalence study | [125] |
| Ropinirol | tablets | BEH C8 (100 × 2.1 mm, 1.7 μm) | A-phosphate buffer, acetonitrile (90:10, $v/v$) B-phosphate buffer, acetonitrile (50:50, $v/v$) Gradient: 0.01/55, 1.7/55, 2.9/98, 3.5/98, 3.6/55, 4.5/55 | 0.27 mL/min | UV 250 nm | degradation study; analysis of impurities; comparison with HPLC | [126] |
| Levetiracetam | blood | BEH C18 (100 × 2.1 mm, 1.7 μm) | A-acetonitrile; B-0.01 M phosphate buffer A:B (10:90, $v/v$) | 0.2 mL/min | UV 215 nm | different ways of the extraction | [127] |
| Piracetam | substance | BEH C18 (150 × 2.1 mm, 1.7 μm) | A-acetonitrile; B-water A:B (25:75, $v/v$) | 0.15 mL/min | UV 210 nm | degradation study; comparison with HPLC | [128] |
| Entacapone | tablets | HSS C18 (50 × 2.1 mm, 1.8 μm) | A-acetonitrile; B-water A:B (43:57, $v/v$) | 0.5 mL/min | UV 225 nm | degradation study; comparison with HPLC | [129] |
| Antidepressants [1] | dosage form | BEH C18 (50 × 2.1 mm, 1.7 μm) | A-acetonitrile; B-10 mM ammonium acetate Gradient: 0/45, 1.75/70, 2.5/80, 3.8/80, 3.9/45, 5/45 | 0.3 mL/min | UV 215 nm | assay | [130] |
| Brexpiprazole | tablets | BEH C18 (50 × 2.1 mm, 1.7 μm) | A-buffer (10 mM $KH_2PO_4$ pH2); B-acetonitrile A:B (67:33, $v/v$) | 0.5 mL/min | UV 215 nm | degradation study | [131] |
| Haloperidol | substance, tablet | CSH fluorophenyl (150 × 2.1 mm, 1.7 μm) | A-0.1% fluoroacetic acid with 10 mM ammonium acetatein water; B-acetonitrile, methanol (80:20, $v/v$) Gradient: 0/20, 15/40, 19.5/20, 23/20 | 0.3 mL/min | UV 246, 220 nm | stability tests; photodegradation study | [132] |

[1] Venlafaxine, Escitalopram, Fluoxetine, Candesartan, Risperidone, Trihexyphenidyl, Thioridazine, Aripiprazole, Trifluoperazine.

## 2.6. Antiviral Drugs

Antiviral drugs are substances that enter cells infected with a virus. They work by inhibiting the attachment of the virus, preventing genetic copying of the virus and the production of viral proteins necessary for its reproduction [133–135]. Viral infections are one of the most common human ailments (causing e.g., colds, flu, warts, etc.); they can also cause infectious diseases, such as HIV/AIDS, Ebola, or COVID-19. Due to the main difference in how a virus replicates in the host cell, there are different classes of antivirals (generally divided into 13 groups). They have been formally approved for the treatment of human infectious diseases such as HIV infection, hepatitis B virus (HBV), HCV, herpes virus, influenza virus, human cytomegalovirus, varicella zoster virus, respiratory syncytial virus, and human papillomavirus. Some examples are acyclovir, zanamivir, and amantadine.

The acyclovir in the cream samples was determined using a Syncronis C18 (100 × 3.0 mm, 1.7 μm) column and a mobile phase with a composition A-0.1 M ammonium acetate buffer and B-acetonitrile, tetrahydrofuran, and water (90:4:6, $v/v/v$); gradient elution: (time [min]/%B set as 0/3, 1.4/3, 2.8/60, 4/60, 4.5/3, 5/3); and a flow rate of 0.5 mL/min. Spectrophotometric detection was performed at a wavelength of 250 nm. It has been proven to be a good method to separate the active ingredient from impurities and

cream ingredients [136]. Vukkum et al. developed a new procedure for the determination of abacavir using the BEH C8 (50 × 2.1 mm, 1.7 μm) column as a stationary phase and a mixture of solvents: A-0.10% o-phosphoric acid in water and B-0.10% o-phosphoric acid in methanol. The mobile phase gradient was used (time [min]T/%B): 0/8, 5/40, 6/40, 6.01/8. The flow rate was 0.4 mL/min, and detection was carried out in the UV range at a wavelength of 220 nm (for the assay of active substance) and with the use of mass spectrometry (for the analysis of impurities) [137]. Another research team performed a determination of five drugs, such as dolutegravir, elvitegravir, raltegravir, nevirapine, and etravirine. The analyzes were carried out with the use of a BEH C18 (50 × 2.1 mm, 1.7 μm) column and a gradient elution (time [min]/%A;B;C: 0/80;20;0, 5/10;90;0, 5.1/10;0;90, 5.9/10;0;90, 6.0/80;20;0, 10.0/80;20;0) of the mobile phase with the following composition: A-0.1% formic acid in water, B-0.1% formic acid in acetonitrile, C-1% formic acid in acetonitrile, with a flow rate of 0.475 mL/min. Eluted components were detected and analyzed using mass spectroscopy [138]. Velpatasvir and sofosbuvir in the form of tablets and substances were analyzed by the UPLC technique using a BEH C18 (150 × 2.1 mm, 1.7 μm) column and a mobile phase containing diammonium phosphate buffer pH6:acetonitrile (40:60, $v/v$). The flow rate was 0.1 mL/min and spectrophotometric detection was set at 280 nm [139]. Another research team performed drug (lamivudine, zidovudine, nevirapine) determination in tablets and substances. The analyzes were carried out with the use of an RP C-18 (100 × 2.1 mm, 1.8 μm) column and isocratic elution of the mobile phase: methanol-phosphate buffer pH5 (70:30, $v/v$) with a flow rate of 1.0 mL/min. Components were detected spectrophotometrically at 260 nm [140].

### 2.7. Antihistamine Drugs

Histamine is a substance that plays an important role in many different body processes, including stimulation of the secretion of gastric acid, dilation of blood vessels, contraction of muscles in the intestines and lungs, and transmission of messages between nerve cells. It is also released if the body encounters an allergen threat. Then it causes the blood vessels to widen, leading to allergy symptoms. Its molecule is an endogenous ligand of histamine receptors, G protein-coupled receptors (GPCR), H1 to H4 [141–143]. Drugs that block the action of histamine, called antihistamines, are generally used to treat histamine-mediated allergic conditions but also anorexia, headaches, anaphylaxis, vertigo, Parkinson's disease (to decrease stiffness and tremors), and some types of bone pain. They are divided into two main groups: the first generation—crossing the blood–brain barrier (e.g., clemastine, and hydroxyzine) and the second generation—not penetrate the blood–brain barrier (e.g., loratadine, cetirizine, and ranitidine). The main metabolite of loratadine, desloratadine, is pharmacologically more potent than the parent compound. It does not easily penetrate the central nervous system readily and therefore has minimal sedative effects.

Desloratadine in the form of a substance and syrup was determined using a BEH C8 (100 × 2.1 mm, 1.7 μm) column and a mobile phase with the composition: A-phosphate buffer, B-acetonitrile:methanol:water (50:25:25, $v/v/v$) (gradient elution, time [min]/%B: 0.0/27, 4.5/32.4, 5.2/80, 5.4/80, 5.5/27, 7.0/27) with a flow rate of 0.4 mL/min. Detections were performed under UV at a wavelength of 272 nm [144]. Rao et al. also conducted studies with desloratadine in the form of tablets. They used a BEH C18 column (50 × 2.1 mm, 1.7 μm) as the mobile phase and a mixture of the following composition as the eluent: A-phosphate buffer:methanol:acetonitrile (80:15:5, $v/v/v$) and B-phosphate buffer:tetrahydrofuran:acetonitrile (30:5:70, $v/v/v$). The mobile phase flow gradient was set to 0.0/0, 1.5/0, 5.5/80, 6.5/80, 7.0/0, 8.0/0 (time [min]/%B). The flow rate was 0.6 mL/min, and the UV detection (at 280 nm) allowed the determination of the active substance content and analysis of the process of its degradation [145]. Dimetindene (as a substance) was analyzed using a BEH C18 (50 × 2.1 mm, 1.7 μm) column and gradient elution (time [min]/%B): 0.0–5.0/95–5, where: A-acetonitrile and B-formic acid. The flow rate was 0.3 mL/mi, and the detection was by mass spectrometry. The conducted research allowed for the analysis of the degradation process of the active substance [146]. Schmidt et al.

determined the ebastine content in the tablets using a BEH C18 column (50 × 2.1 mm, 1.7 μm) and a mixture of A-10 mM acetate buffer and B-acetonitrile:2-propanol (1:1, *v/v*) as the mobile phase. The elution rate of 0.5 mL/min was carried out with the gradient: 0.0–3.0/30–90 (time [min]/%B) and the UV detection at λ = 210 nm [147]. The mixture of ambroxol and cetyrizine in the form of a tablet and oral solution was analyzed on an Agilent Eclipse plus C18 column (50 × 2.1 mm, 1.8 μm) with a mixture of solutions 0.01 M phosphate buffer (A) and 0.1% trimethylamine in acetonitrile (B). The gradient elution was carried out according to the program (time [min]/%B): 0.0/30, 0.2/30, 3.0/95, 3.1/30, 3.5/30. The flow rate was 0.5 mL/min. Spectrophotometric detection in UV at a wavelength of 237 nm allowed for the quantitative analysis of the active substance [148]. Chambers et al. presented the analytical procedure for the determination of ibuprofen, pseudoephedrine, and chlorpheniramine in tablets. They used an Acquity BEH C18 (50 × 2.1 mm, 1.7 μm) column as a stationary phase and a mixture of A (0.1% triethylamine buffer pH3.2 with phosphoric acid and acetonitrile (80:20, *v/v*)) and B (0.1% triethylamine buffer pH3.2 with phosphoric acid and acetonitrile (50:50, *v/v*)) as the eluent. Gradient conditions were as follows: 1/5, 2/5–80, 1/80 (time [min]/%B). The determined value of the flow rate was 0.4 mL/min and UV detection was at 220 nm. The developed conditions made it possible to carry out a degradation study of the active substance [70].

### 2.8. Other Drugs

In the group of active substances presented in this subchapter, there are active substances from different therapeutic groups [65,96]. A large group consists of compounds influencing the hormonal balance, used both in hormone replacement therapy, e.g., gestodene, estradiol, and in anti-cancer treatment (e.g., abiraterone, finasteride). Another part consists of substances used in the treatment of hyperglycemia (sitagliptin and metformin) and anticancer drugs, both classic cytotoxic drugs (topotecan) and targeted drugs (imatinib). Preparations used in lung diseases constitute a large group of drugs that were not considered before. These include inhaled β2 mimetics (salbutamol and fenoterol) as well as other asthma medications, e.g., montelukast. In addition to those mentioned, there are also such drugs as lansoprazole and omeprazole, used in the treatment of peptic ulcer disease; tramadol, a strong pain reliever from the opioid group; and tolterodine and darifenacin, used mainly in the treatment of urinary incontinence. The parameters of the UPLC analysis used to research these drugs are presented in Table 4.

**Table 4.** UPLC technique in the analysis of other drugs from various therapeutic groups.

| Active Substance | Sample | Column | Mobile Phase (Gradient: Time [min]/%B) | Flow Rate | Detection | Comments | Ref |
|---|---|---|---|---|---|---|---|
| Tramadol | solution for injections | BEH C18 (100 × 2.1 mm, 1.7 μm) | A-0.2% trifluoroacetic acid buffer B-methanol, acetonitrile (75:25, *v/v*) Gradient: 0/20, 15/60, 16/20, 20/20 | 0.2 mL/min | UV 275 nm | stability test after reconstitution in saline and glucose | [149] |
| Lansoprazole | capsules, suspensions | BEH C18 (100 × 2.1 mm, 1.7 μm) | A-water; B-acetonitrile with 0.1% formic acid A:B (60:40 *v/v*) | 0.2 mL/min | MS, TOF-MS | stability testing | [150] |
| Imatynib | plasma | BEH Shield RP18 (50 × 2.1 mm, 1.7 μm) | A-ammonium formate in waterB-acetonitrile, 0.1% formic acid Gradient: 0/2, 0.5/2, 0.5–2.5/2–50, 2.5–3/50–90,3–4.5/90 | 0.4 mL/min | MS/MS | assay | [151] |
| Clenbuterol Terbutalin Salbutamol Fenoterol Genistein Daidzein Tamoxifen Ephedrine Pseudoephedrine | substance | Acquity RP (50 × 1.0 mm, 1.7 μm) | A-acetonitrile; B-0.1% formic acid A:B (40:60, *v/v*) | 0.2 mL/min | MS | comparison with HPLC | [152] |
| Amphetamine Methamphetamine | urine | BEH C18 (50 × 2.1 mm, 1.7 μm) | A-ammonium formate; B-methanol Gradient: 0–0.15/5, 0.15–0.3/5–30, 0.3–2/30–40, 2–3/40–50, 3–4.2/50–98, 4.2–5.2/98, 5.2–5.4/98–5, 5.4–5.8/5 | 0.4 mL/min | MS/MS | assay | [153] |
| Azathioprine | substance | BEH C18 (100 × 2.1 mm, 1.7 μm) | A-0.05% trifluoroacetic acid in water; B-acetonitrile Gradient: 0/3, 1/3, 3.5/60, 4/60, 4.1/3, 5/3 | 0.35 mL/min | UV 220 nm | assay | [154] |
| Ranitidine | substance | BEH C18, C8, phenyl, C18 Shield (100 × 2.1 mm, 1.7 μm) | A-ammonium bicarbonate; B-methanolGradient: 0/4, 1/16, 4/36, 7/90 | 0.45 mL/min | UV 230 nm; MS | degradation study; comparison of different columns and eluents; comparison with HPLC | [155] |
| Dienogest Finasterid Gestodene Levonorgestrel Estradiol Ethinylestradiol | substance | BEH C18 (50 × 2.1 mm, 1.7 μm) | A-acetonitrile; B-water; A:B (48:52, *v/v*) | 0.55 mL/min | UV 210 nm | purity testing | [156] |
| Caffeine Theobromine Theophilline | tablets | BEH C18 (2.1 × 50 mm, 1.7 μm) | A-ammonium acetate; B-acetonitrile Gradient: 0–1/5, 2–2.5/20, 3–3.5/80 | 0.6 mL/min | UV 275 nm | assay in dietary supplements | [157] |
| Dantrolen | substance | BEH C18 (50 × 2.1 mm, 1.7 μm) | A-2.5 mM sodium acetate buffer pH4.5 B-acetonitrile; A:B (75:25, *v/v*) | 0.5 mL/min | UV 375 nm; MS; NMR | degradation study | [158] |
| Mesalazine | tablets | BEH C18 (50 × 2.1 mm, 1.7 μm) | A-buffer pH2.2; B-buffer pH6, methanol, acetonitrile (890:80:30, *v/v/v*) Gradient: 0/10, 3/10, 13/90, 13.1/10, 15/10 | 0.7 mL/min | UV 220 nm | assay | [159] |
| Sitagliptine Metformin | combined tablets | BEH C8 (100 × 2.1 mm, 1.7 μm) | A-phosphoric acid; B-acetonitrile Gradient: 0/8.0, 2/8.0, 4/45, 6/45, 8/8, 10/8 | 0.2 mL/min | UV 210 nm | assay | [160] |
| Ranolazine | tablets | BEH RP18 (100 × 2.1 mm, 1.7 μm) | A-acetonitrile, phosphate buffer pH7.3, triethylamine (10:90:0.1, *v/v/v*) B-acetonitrile, phase A (55:45, *v/v*) Gradient: 0.01/17, 1.5/17, 3.5/45, 5.5/60, 8/65, 12/70, 13/95, 15/95, 15.5/17, 18/17 | 0.3 mL/min | UV 223 nm | degradation study | [161] |
| Darifenacin | tablets | BEH C18 (100 × 2.1 mm, 1.7 μm) | A-triethylamine + phosphate buffer (1:1000, *v/v*), acetonitrile (80:20, *v/v*) B-triethylamine + phosphate buffer (1:1000, *v/v*), acetonitrile (15:85, *v/v*) Gradient: 0/15, 2/15, 10/50, 14/74, 14.1/15, 15/15 | 0.3 mL/min | UV 210 nm | assay | [162] |
| Cyanocobalamin (vitamin B12) | substance | BEH C18 (50 × 1.0 mm, 1.7 μm) | A-0.1% trifluoroacetic acid in water B-0.1% trifluoroacetic acid in acetonitrile Gradient: 0–0.25/5, 0.25–2.5/5–40, 2.5–3/40, 3–3.5/40–5 | 0.32 mL/min | UV 254 nm | assay | [163] |
| Bicalutamide | tablets, substance | HSS T3 (100 × 2.1 mm, 1.8 μm) | A-0.001 M sodium dihydrogen orthophosphate pH6 with sodium hydroxide B-acetonitrile, phase A (90:10, *v/v*) Gradient: 0/28, 26/55, 29.3/55, 31.3/28, 34/28 | 0.5 mL/min | UV 220 nm | degradation study; analysis of impurities | [164] |
| Imatinib | tablets | BEH C18 (50 × 2.1 mm, 1.7 μm) | A-0.05 M ammonium acetate pH9.5 B-acetonitrile, methanol (40:60, *v/v*) Gradient: 0.01/42, 5/42, 7/80, 8/42, 9/42 | 0.3 mL/min | UV 237 nm | degradation study | [165] |

**Table 4.** *Cont.*

| Active Substance | Sample | Column | Mobile Phase (Gradient: Time [min]/%B) | Flow Rate | Detection | Comments | Ref |
|---|---|---|---|---|---|---|---|
| Triamcinolone Hydrocortisone Indometacin Etradiol | creams, gels | BEH C18 (2.1 × 50 mm, 1.7 μm) | A-acetonitrile; B-water; A:B (40:60, *v/v*) | 0.6 mL/min | UV 240 nm | assay | [166] |
| Lanzoprasole | tablets | BEH-C18 (50 × 2.1 mm, 1.7 μm) | A-8 mL triethylamine in 20 mM $KH_2PO_4$ buffer pH7 with orthophosphoric acid, methanol (90:10, *v/v/v*) B-methanol, acetonitrile (50:50, *v/v*) Gradient: 0.01/20, 2/30, 5/50, 6/70, 8.5/70, 9.5/20, 11/20 | 0.3 mL/min | UV 285 nm | degradation study | [167] |
| Erythropoietin | substance | BEH C18 (50 × 2.1 mm; 1.7 μm) | A-0.1% trifluoroacetic acid in water B-0.1% trifluoroacetic acid in acetonitrile Gradient: 0/15, 0.12/15, 0.33/30, 0.62/36, 2.62/65, 3.19/100, 3.76/15, 4.05/15 | 0.35 mL/min | UV 210 nm | assay in human serum albumin; comparison with HPLC | [168] |
| Topotecan | solution for injections, substance | BEH C18 (50 × 2.1 mm, 1.7 μm) | A-0.1% orthophosphoric acid in water B-acetonitrile Gradient: 0/10, 0.5/10, 1/20, 2/20, 3/10, 4/10 | 0.4 mL/min | UV 260 nm | assay | [169] |
| Bortezomib | substance | ULTRAFAST Shimpack XR-ODS-II (100 × 3 mm, 2.2 μm) | A-potassium dihydrogen phosphate buffer B-acetonitrile Gradient: 0/20, 2/30, 5/50, 6/70, 8/20, 10/20 | 0.6 mL/min | UV 270 nm; MS | analysis of impurities | [170] |
| Tramadol | tablets | BEH C18 (100 × 2.1 mm, 1.7 μm) | A-potassium dihydrogen phosphate buffer B-acetonitrile; A:B (60:40 *v/v*) | 0.5 mL/min | UV 226 nm | degradation study | [171] |
| Tolterodine | tablets, serum, urine | BEH C18 (100 × 2.1 mm, 1.7 μm) | A-0.025% trifluoroacetic acid in water B-0.025% trifluoroacetic acid in acetonitryl Gradient: 0/30, 4/80, 6/80, 6.1/30 | 0.3 mL/min | UV 220 nm | assay | [172] |
| Bambuterol Montelukast | tablets | BEH C18 (100 × 2.1 mm, 1.7 μm) | A-0.025% trifluoroacetic acid in water B-0.025% trifluoroacetic acid in acetonitrile Gradient: 0/30, 1.5/40, 3/90, 6/90, 6.1/30 | 0.3 mL/min | UV 210 nm | assay | [173] |
| Uracil Chlorphromazine Imipramine Clozapin Diltiazem Bifonazole | substance | BEH C18 (50 × 2.1 mm, 1.7 μm) | A-0,1% formic acid in water; B-acetonitrile Gradient: 0/5, 1/90, 1.1/5, 2/5 | 0.3 mL/min | MS | solubility testing in various media; comparison with HPLC | [174] |
| Terbutaline | substance | Phenomenex luna C18 (150 × 2.0 mm, 3 μm) | A-ammonium formate buffer; B-methanol Gradient: 0–6/5, 6–15/5–30, 15–20/30–80, 20–23/80–90, 23–23.1/90–5 | 0.3 mL/min | QTOF-MS | degradation study; *in silico* toxicity tests | [175] |
| Esomeprazole | plasma | BEH C18 (50 × 2.1 mm, 1.7 μm) | A-acetonitrile with 0,1% formic acid B-ammonium formate with water Gradient: 0–0.7/80, 0.8–1.7/80–20, 1.8–2.3/20, 2.4–3/20 | 0.4 mL/min | QTOF-MS | pharmacokinetics study | [176] |
| Abiraterone Letrozole Anastrozole Bicalutamid | plasma | BEH C18 (50 × 2.1 mm, 1.7 μm) | A-0.1% formic acid in water B-acetonitrile, methanol (50:50, *v/v*) Gradient: 0–4/45, 4–5/100, 5–6/45 | 0.6 mL/min | MS | assay | [177] |
| Pseudoephedrine Chlorpheniramine Ibuprofen | tablet | Acquity BEH (50 × 2.1 mm, 1.7 μm) | A-0.1% formic acid in water B-0.1% formic acid in methanol Gradient: 1/5, 2/5–80, 1/80 | 0.3 mL/min | MS | assay | [87] |
| Hydrocortisone Tinidazole | substance, vaginal tablet, cream | Acquity Eclipse plus C18 (100 × 2.1 mm, 1.7 μm) | A-0.02 M anhydrous $KH_2PO_4$ (with 0.2% triethylamine) pH6 with orthophosphoric acid B-acetonitrile Gradient: 0/50, 2/70, 5.6/70, 5.7/50, 7/50 | 0.3 mL/min | UV 225, 295 nm | determination of impurities | [106] |

**Table 4.** *Cont.*

| Active Substance | Sample | Column | Mobile Phase (Gradient: Time [min]/%B) | Flow Rate | Detection | Comments | Ref |
|---|---|---|---|---|---|---|---|
| Lenvatinib Telmisartan | substance, plasma | X Select HSS T3 (100 × 2.1 mm, 2.5 μm) | A-water with 0.1% formic acid and 5 mM ammonium acetate<br>B-acetonitrile with 0.1% formic acid<br>Gradient: 2/60, 2–3/60–90, 3–4/90, 4–4.1/910–60, 4.1–5.1/60 | 0.25 mL/min | MS-MS | assay | [64] |
| Venetoclax | human plasma | Acquity BEH (100 × 2.1 mm, 1.8 μm) | A-0.1% formic acid in water; B-acetonitrile<br>Gradient: 0–0.3/5, 0.3–2/5–95, 2–2.5/95, 2.5–2.6/95–5, 2.6–4/5 | 0.4 mL/min | MS-MS | assay | [178] |
| Actinomycin D | substance, brain tissue, plasma | Peptide C18 (50 × 2.1 mm, 1.7 μm) | A-5% acetonitrile in water with 0.1% formic acid<br>B-acetonitrile with 0.1% formic acid<br>Gradient: 0–0.5/40, 0.5–2/40–100 | 0.5 mL/min | MS-MS | microdialysis model | [179] |
| Famotidine Ibuprofen | tablet | Acquity BEH C-18 (50 × 2.1 mm, 1.7 μm) | A-50 mM sodium acetate buffer pH5.5<br>B-methanol; A:B (25:75, *v/v*) | 0.3 mL/min | UV 260 nm | assay | [91] |
| Glucagon | for injection | Acquity BEH 300 C-18 (100 × 2.1 mm, 1.7 μm) | A-phosphate buffer pH2.7 (with phosphoric acid)<br>B-acetonitrile, water (4:6, *v/v*); A:B (65:35, *v/v*) | 0.4 mL/min | UV 214 nm | stability study | [180] |
| Lansoprazole Naproxen | substance, tablet | Phenomenex Luna C18 (250 × 4.6 mm, 5 μm) | A-methanol; B-water; A:B (8:2, *v/v*) | 1.0 mL/min | PDA | assay | [92] |
| Glucocorticoids [1] Clobetasol Beclomethasone Flucinonide Desonide | tablet | HSS T3 (100 × 2.1 mm, 1.8 μm) | A-0.1% formic acid with 5 mM ammonium formate in water; B-0.1% formic acid in acetonitrile<br>Gradient: 0–10/30–95, 10–15/95 | 0.2 mL/min | QTOF-MS | determination in dietary supplements | [181] |
| Cathinones [2] Opiates Cocaine/related compounds Scopolamine | oral fluid | Acquity BEH Shield RP18 (100 × 2.1 mm, 1.7 μm) | A-0.1% formic acid in water<br>B-0.1% formic acid in acetonitrile<br>Gradient: 0–0.2/10, 3.5/70, 4/10 | 0.4 mL/min | MS-MS | assay | [182] |

[1] Prednisolone, Prednisone, Riamocinolone acetonide, Dexamethasone, Hydrocortisone, Cortisone. [2] Morphine, 6-Monoacetylmorphine, Cocaine, Cocaethylene, Benzoylecgonine, Methadone, Methylenedioxypyrovalerone, Mephedrone, Methylone, Buprenorphine, Naloxone, Pentedrone, Ethylone, Butylone, Ethylcathinone, Ethylcathinone ephedrine metabolite, Methylephedrine metabolite, Pyrovalerone, Flephedrone, Scopolamine.

### 2.9. Summary

It can be seen that the UPLC technique is now an increasingly used tool in the analysis of drugs. It allows the identification of various chemical components and the determination of their content, which translates into a wide range of applications in scientific research and production. This article provides a general overview of medically important drugs and their analysis by UPLC. The above UPLC applications in the analysis of pharmaceutical substances focused on compounds with biological activity belonging to various pharmacological groups commonly used in medicine. This work focuses on the characteristics of the systems used in the analysis of active substances in drugs, also in the presence of other co-existing ingredients.

UPLC methods are used to separate mixtures and identify many chemical compounds (in addition to those listed above, also amino acids, nucleic acids, proteins, steroids, etc.) [183], check the purity of manufactured drugs to ensure product quality [73], monitor the kinetics of chemical reactions (including the synthesis of new structures of potential therapeutic importance) [184], study physicochemical properties, i.e., lipophilicity (alongside the commonly used TLC method) [185,186], perform isomer analysis [187], or complete stability tests in changing environmental conditions [188]. Maximized pressure, minimal lag, and fast injections enable very fast cycle times while maximizing peak yields. For example, when assaying a combination tablet containing diclofenac, paracetamol, and camylofin, the UPLC analysis time was shown to be four times shorter compared to HPLC, and solvent consumption was approximately sixteen times lower [83]. The results of HPLC and UPLC analyses for piracetam were also compared, recording 10 times lower LOD and LOQ values for the same assay in favor of UPLC and a six-times shorter analysis time in isocratic mode [128]. By using UPLC instead of HPLC for the determination of erythropoietin, the total analysis time was reduced from 20 to 4 min while obtaining a greater range of linearity of the method [168]. Precisely because of the speed, resolution, and sensitivity of the apparatus, UPLC methods are very well suited for use with a mass spectrometer, which increases the possibilities of this technique and makes it a practical and reliable tool for more laboratories, allowing for precise solvent administration, perfect reproducibility, and minimal sample transfer.

With the growing need for accurate measurements to support drug discovery and further development, the demand for selective and sensitive chromatographic methods has significantly increased. Although quantification by HPLC has many advantages, these analyses pose many challenges for technical development related to the insufficient recovery of components after extraction (new requirements for sample preparation). The features of the UPLC technique, i.e., increased analytical sensitivity, linear dynamic range, or high repeatability, enable the measurement of low concentrations of ingredients, demonstrating its suitability for the purposes of discovering new drugs and quality control of raw materials and products as well as clinical trials.

The main advantages of UPLC (i.e., shortening the analysis time and reducing the volume of the mobile phase) indicate a great development possibilities of this technique. Compared to HPLC-based methods, UPLC, thanks to better chromatographic resolution (ensures the elimination of the potential influence of a complicated matrix), increased sensitivity, and shorter analysis times, reduces the cost and increases the efficiency of the analysis required to develop and validate the method. The list of developed protocols is a contribution to the existing trend and limitations in this area of research. According to the assumptions of 'green chemistry', better and better solutions for drug analysis should be sought, e.g., by searching for less toxic solvents (characterized by high viscosity, high thermal stability, and low vapor pressure). Problems may include high pressure or insufficient quality of solvents. To avoid these complications, particular attention must be paid to the temperatures of the dispenser, filters, and pumps or rotating loops, etc. It is especially important to regularly clean the entire system.

Future trends in drug analysis aim at minimizing both the size of chromatographic columns and their fillings, increasing the resolution and sensitivity of detection as well as minimizing the time and cost of these tests. Transferring the conditions from HPLC to UPLC is not difficult, but there are a few issues to consider, generally related to instrumentation requirements (to achieve higher pressures and maintain accuracy and precision at lower flow rates, higher capacity pumps and components are required). Moreover, such a procedure requires time and resources to optimize. Our work is intended to be a source of such information about the already adapted methodology of assays. This knowledge can complement the drug information database and storage guidelines, increasing the number of tools for quality control and safer treatments.

## 3. Conclusions

As we can see, the UPLC technique is already an established and rapidly developing field with many possible applications in the analysis of pharmaceutical substances. The use of the UPLC technique for the analysis of medicinal substances in various pharmaceutical products presented in this manuscript indicates the great importance of this technique in the analysis of drugs while also showing the problems that can occur when adapting the method conditions from the HPLC system to the UPLC. This topic may turn out to be even more important when analyzing medicinal substances in a more complex matrix, i.e., biological material. The presented conditions of analytical procedures using the UPLC technique confirm its advantages, such as high resolution, sensitivity, and shorter analysis time. Thus, transferring the legacy conditions of the HPLC method to the UPLC may be a beneficial process. The presented data show that UPLC can become a basic tool of an analyst's work to improve the quality of pharmaceutical analysis and research capabilities. As you know, the more information we have about a given active substance (its quality, stability, interaction with other substances, etc.), the effectiveness of the therapy in which a drug containing this substance is used will be more effective.

**Author Contributions:** Conceptualization, M.S. and P.G.; methodology, P.G. and J.Ż.; validation, M.D. and M.S.; investigation, P.G. and J.Ż.; writing—original draft preparation, P.G. and M.S.; writing—review and editing, M.S. and M.D.; visualization, M.S.; supervision, M.S. All authors have read and agreed to the published version of the manuscript.

**Funding:** This research received no external funding.

**Data Availability Statement:** Not applicable.

**Conflicts of Interest:** The authors declare no conflict of interest.

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
