# Peer review of "UPLC Technique in Pharmacy—An Important Tool of the Modern Analyst"

_processes, doi:10.3390/pr10122498_

Round 1
Reviewer 1 Report
The manuscript “UPLC technique in pharmacy – an important tool of the modern analyst” wrote by PaweÅ‚ GumuÅ‚ka et al. collected the latest applications of the UPLC analytical method in drug quality analysis. Four tables were listed to show the applications of UPLC technique in the analysis of drugs. The article is written carefully. However, most part of the manuscript is relevant to the introduction of the drugs instead of the discussion of the applications of the UPLC analytical method (for example: section 2.1, section 2.2, section 2.3, section 2.5, section 2.8). The discussion digressed from the subject.
Author Response
-

Reviewer 2 Report
The proposed review gives an overview of the important topic from a pharmaceutical analysis point of view.
The main text is clearly written. From my perspective, tables need some editorial efforts. Additionally, considering the importance of lipophilicity from a pharmaceutical chemistry point of view, I believe this topic should be mentioned in the form of two-three sentences (https://doi.org/10.1016/j.chroma.2017.09.015), and some examples should be given https://doi.org/10.1007/s00216-009-2862-1, together with advantages of UPLC.
Author Response
-

Reviewer 3 Report
Dear Author, I reviewed the manuscript (processes-2004757) entitled UPLC technique in pharmacy – an important tool of the modern analyst. This manuscript presents relevant information about UPLC chromatographic techniques applications in the pharmaceutical area. However, some sections of the presented data can be improved. For this reason, I consider that this manuscript needs minor changes to be considered for publication in this journal.
Additional comments.
Highlight the advantages of using UPLC protocols compared with other conventional chromatographic tools.
Check the paragraph extension in this manuscript.
Try to compare the obtained findings with other analytic protocols to evaluate antibiotics applied in pharmacy.
Include future trends to keep working with the obtained data.
Try to conclude with a general statement of the most relevant part of this study.
Author Response
-

Round 2
Reviewer 1 Report
I think the manuscript has been sufficiently improved to warrant publication in Processes now.